# Amblyopia risk factors among pediatric patients in a hospital-based setting using photoscreening

Christiane Al-Haddad[1]*, Zeinab El Moussawi[1], Stephanie Hoyeck[1], Carl-Joe Mehanna[1], Nasrine Anais El Salloukh[1], Karine Ismail[1], Mona Hnaini[2], Rose-Mary N. Boustany[2]

1 Ophthalmology Department, American University of Beirut, Beirut, Lebanon, 2 Pediatric Neurology Division, Department of Pediatrics and Adolescent Medicine, AUBMC Special Kids Clinic, Beirut, Lebanon

* ca12@aub.edu.lb

**Data Availability Statement:** All relevant data are within the paper and its Supporting information files.

## Abstract

### Purpose

The aim of our study was to determine the prevalence of amblyopia risk factors in children visiting the American University of Beirut Medical Center (AUBMC) using automated vision screening.

### Methods

This was a hospital-based screening of 1102 children aged between 2 and 6 years. Vision screening was performed using PlusoptiX S12 over 2 years (2018–2020). The need for referral to a pediatric ophthalmologist was based on the amblyopia risk factors set forth by the American Association for Pediatric Ophthalmology and Strabismus. Referred patients underwent a comprehensive eye examination.

### Results

A total of 1102 children were screened, 63 were referred for amblyopia risk factors (5.7%); 37/63 (59%) underwent comprehensive eye examination and 73% were prescribed glasses. Of the non-referred group of children, 6.35% had astigmatism, 6.25% were hyperopic and 3.27% were myopic. The refractive errors observed among the examined patients were distributed as follows: 41% astigmatism, 51% hyperopia, and 8% myopia; amblyopia was not detected. Refractive amblyopia risk factors were associated with the presence of systemic disorders. Bland-Altman plots showed most of the differences to be within limits of agreement.

### Conclusion

Using an automated vision screener in a hospital-based cohort of children aged 2 to 6 years, the rate of refractive amblyopia risk factors was 5.7%. Hyperopia was the most commonly encountered refractive error and children with systemic disorders were at higher risk.

**Funding:** This study was funded by the Medical Practice Plan (MPP) of the American University of Beirut awarded to CA (https://www.aub.edu.lb/fm/medicalresearch/Pages/ResearchFunding.aspx). The funders had no role in study design, data collection and analysis, decision to publish, or preparation of the manuscript.

**Competing interests:** The authors have declared that no competing interests exist.

## Introduction

Early childhood vision screening is recommended for detecting preventable and treatable vision disorders [1–3]. Amblyopia is a common cause of decreased vision in children [4–11]. Early diagnosis through vision screening, referrals for complementary examinations and early interventions lead to better visual outcomes [4]. The US Preventive Services Task Force recommends vision screening at least once in all children aged 3 to 5 years to detect amblyopia or its risk factors [2]. The American Academy of Ophthalmology (AAO) and the American Association for Pediatric Ophthalmology and Strabismus (AAPOS) [12] recommend vision screening during the preschool years [3]. Instrument-based screening is recommended for all children aged 12 months and above undergo [3, 13]. In one study, photoscreening was superior to optotype-based screening for children between the ages of 3 and 6 years [14]. In addition, children screened and referred before the age of 2 years were more likely to reach a visual acuity of 20/40, as compared to those screened after age 2 [15].

One of the recent automated screeners is PlusoptiX; it is a digital portable photoscreener that gives a non-cycloplegic autorefraction of undilated pupils [16]. PlusoptiX has shown a sensitivity ranging from 92.86% to 100% and a specificity ranging from 49.57% to 94.49% in the detection of refractive amblyopia risk factors [17]. A recent study compared three photoscreening devices, iScreen, PlusoptiX and SPOT photoscreeners: sensitivity overall for detection of amblyogenic factors ranged from 72% (iScreen) to 84% (PlusoptiX), and specificity ranged from 68% (SPOT) to 94% (PlusoptiX) [18].

In studies from several parts of the world, the prevalence of refractive errors in children showed a wide range [17, 19]. The subtypes of refractive risk factors were: 3.5% to 30.5% for hyperopia, 4.9% to 53.1% for myopia and 22% to 25.5% for astigmatism [19–25].

There is scarce literature on the prevalence of refractive amblyopia risk factors in the Middle East, and more specifically in Lebanon. The aim of our study was to detect refractive amblyopia risk factors in children visiting the American University of Beirut Medical Center for a routine check-up at the pediatrics clinics, using the PlusoptiX S12 photoscreener, and to correlate results with age, sex, systemic disorders and prematurity.

## Methods

### Population

This was a hospital-based screening of 1102 children, aged 2 to 6 years, visiting the general pediatric clinics of the American University of Beirut Medical Center (Out-Patient Division and private clinics). The study was approved by the Institutional Review Board (IRB) at the American University of Beirut Medical Center (IRB ID: OPH.CH.20) and was conducted over a 2 year period from 2018 to 2020. Written informed consent was obtained from parents or legal guardians. The study adhered to the principles of the Declaration of Helsinki. Demographic data were collected from the parents, including date of birth, prematurity, presence of any known eye disease, and past medical history of any systemic disease.

### Vision screening

Vision screening was performed by personnel who had received a three-hour-training session on the use of the PlusoptiX S12 photoscreener (PlusoptiX GmbH, Atlanta, GA). Performing vision screening was simple: the camera was set off by pulling the trigger. The sound attracted attention and both eyes were captured on screen in a white rectangle and measurement was automatically registered. Screening results were immediately displayed on the screen and data on refraction (sphere, cylinder and axis) of each eye were recorded. Spherical equivalent (SE)

**Table 1. AAPOS amblyopia risk factors targeted with automated preschool vision screening [26].**

| Age, months | Astigmatism | Hyperopia | Anisometropia | Myopia |
|---|---|---|---|---|
| 12–30 | >2.00D | >4.50D | >2.50D | > -3.50D |
| 31–48 | >2.00D | >4.00D | >2.00D | > -3.00D |
| >48 | >1.50D | >3.50D | >1.50D | > -1.50D |

D: Diopters

refraction in diopters (D) was then calculated. If the primary screening attempt failed, screening was repeated up to three times.

If the screening revealed refractive errors, subjects were referred to the pediatric ophthalmology clinic (author CA) according to the age-based criteria for amblyopia risk factor detection guidelines set by the American Association of Pediatric Ophthalmology and Strabismus Vision Screening Committee (Table 1) [26].

## Eye examination

To confirm the results of the vision screening when positive, a comprehensive pediatric eye examination was performed by the same attending pediatric ophthalmologist (author CA). This entailed age-dependent visual acuity testing: "central steady & maintained" testing for preverbal children and vision charts (Allen pictures and Early Treatment Diabetic Retinopathy Study charts (EDTRS)) for verbal children. Anterior segment examination, motility examination for any eye misalignment, and posterior segment examination using indirect ophthalmoscopy were performed. Cycloplegic manual retinoscopy was done to detect refractive errors (30 minutes after pupillary dilation with Mydriacyl 1% and Cyclopentolate 1%, applied twice 10 minutes apart; in this clinic, we use two sets as a standard especially with darker irides). Significant astigmatism was defined as cylindrical power of 1D or more with the main refractive error being the cylinder. Those reported with hyperopia and myopia had mainly spherical refractive error (>+1D or <-1D, respectively).

## Statistical analysis

After obtaining complete data, statistical analysis was performed using SPSS (Statistical Packages for Social Sciences) v 24, where frequencies and descriptive statistics were computed. Correlations were established using independent t-tests, bivariate (Pearson) correlation, intraclass correlation coefficient (ICC) score. Significance level was set at a p-value of less than 0.05. Bland-Altman analysis was performed by calculating the difference between spherical equivalent values obtained by PlusoptiX and by cycloplegic refraction for each eye, plotted against the average of the two values. Horizontal dashed lines were drawn at the 95% limits of agreement, which were defined as the mean difference ±1.96 x standard deviation of the differences.

## Results

### Characteristics of study population

A total of 1102 children aged 2 to 6 years were screened using the PlusoptiX S12 vision screener. Mean age of the studied population was 4.13 ± 1.44 years, with 511 females (46%). Twenty-one of included subjects (1.9%) were born prematurely (gestational age below 37 weeks), and 93 (8.4%) had associated systemic disorders. The mean SE refraction was 0.39 ± 0.74D for right eyes, and 0.40 ± 0.71D for left eyes. The prevalence of refractive

**Table 2. Demographics of the studied population compared to the demographics of the population subgroup requiring referral to a comprehensive eye exam.**

|  | Total | Referred | P-value* |
|---|---|---|---|
|  | N = 1102 | N = 63 |  |
| Mean age ± SD (years) | 4.13±1.44 | 4.24±1.48 | 0.63 |
| Gender Female, n (%) | 511 (46%) | 28 (44.5%) | 0.76 |
| Premature, n (%) | 21 (1.9%) | 1 (1.6%) | 0.86 |
| Systemic disorders, n (%) | 93 (8.4%) | 14 (22.2%) | **< 0.001** |
| Hematologic | 22 | 4 |  |
| Neurologic | 22 | 3 |  |
| Cardiac | 15 | 4 |  |
| Metabolic | 15 | 2 |  |
| Respiratory | 12 | 1 |  |
| Immunologic/Rheumatologic | 5 | 0 |  |
| Syndromic | 2 | 0 |  |
| Range of spherical equivalent (in D) |  |  |  |
| OD | (-3.88)–(+4.50) |  |  |
| OS | (-3.88)–(+3.75) |  |  |
| Mean spherical equivalent refraction ± SD (in D) |  |  |  |
| OD | 0.39 ± 0.74 | 0.75 ± 1.68 | 0.15 |
| OS | 0.40 ± 0.71 | 0.74 ± 1.67 | 0.15 |

*p-values comparing the referred group to the total

amblyopia risk factors was 5.7%, where 63 subjects were referred for full eye examination according to the AAPOS age-appropriate refractive error targets (Table 1).

Among the 63 subjects who met the criteria for referral, 37 (59%) were examined at the pediatric ophthalmology clinic of our department (by author CA). The rest (41%) were lost to follow up after multiple attempts to schedule an ophthalmology appointment.

Mean age of the referred group was 4.24±1.48 years, with no statistically significant difference compared to that of the whole population studied: 28 were females (44.5%),1 was premature (1.6%) and 14 had associated systemic disorders (22.2%). According to the screening exam with the automated vision screener, mean SE was 0.75 ± 1.68D for right eyes, and 0.74 ± 1.67D for left eyes, similar to those for the whole group (Table 2). After cycloplegic refraction for this group, the detected refractive errors were as follows: 41% astigmatism (mean cylinder 2.04±0.85D), 51% hyperopia (mean SE +3.28±2.79D), and 8% myopia (mean SE -2.25±1.89D). Of the examined subjects, 27 (73%) were prescribed glasses. Of the non-referred group of children with minimal refractive errors not reaching the AAPOS threshold values, automated vision screening revealed astigmatism in 66 (6.35%) with mean cylinder of 1.24 ± 0.27D, hyperopia in 65 (6.25%) with mean SE of +1.31 ± 0.42D and myopia in 34 (3.27%) with mean SE of -1.42 ± 0.44D.

## Comparison between the plusoptiX and cycloplegic retinoscopy

The plusoptiX and CR results are compared in Table 3. For the whole examined population (n = 37), a significant difference was seen for the sphere (p<0.001) and the mean SE (p = 0.02), with a difference of 1.21 ± 2.61D and 0.54 ± 2.48D, respectively. The intraclass correlation coefficient (ICC) score of the SE between Plusoptix and cycloplegic refraction was 0.70. Out of the 37 examined children, 19 had hyperopia, 15 had astigmatism and 3 subjects were myopic. When looking at each refractive error separately, the difference between cycloplegic refraction

**Table 3. Comparison between the plusoptiX and cycloplegic retinoscopy of the examined population.**

| | Sphere (D) | | Cylinder (D) | | SE (D) | |
|---|---|---|---|---|---|---|
| | Mean ± SD | p-values | Mean ± SD | p-values | Mean ± SD | p-values |
| Total (n = 37) | | | | | | |
| CRx | 1.68 ± 2.75 | <0.001 | 1.31 ± 1.07 | 0.27 | 1.47 ± 2.40 | 0.023 |
| PlusoptiX | 0.47 ± 1.27 | | 1.44 ± 0.99 | | 0.93 ± 1.57 | |
| Difference | 1.21 ± 2.61 | | 0.13 ± 0.98 | | 0.54 ± 2.48 | |
| Astigmatism (n = 15) | | | | | | |
| CRx | 0.75 ± 0.84 | 0.14 | 2.04 ± 0.85 | 0.80 | 1.77 ± 0.99 | 0.18 |
| PlusoptiX | 0.30 ± 0.80 | | 1.96 ± 0.90 | | 1.28 ± 1.00 | |
| Difference | 0.45 ± 0.04 | | 0.08 ± 0.05 | | 0.49 ± 0.01 | |
| Hyperopia (n = 19) | | | | | | |
| CRx | 3.11 ± 2.70 | 0.0026 | 0.81 ± 0.90 | 0.21 | 3.28 ± 2.80 | 0.01 |
| PlusoptiX | 0.85 ± 1.40 | | 1.18 ± 0.90 | | 1.44 ± 1.40 | |
| Difference | 2.26 ± 1.30 | | 0.37 ± 0.00 | | 1.84 ± 1.40 | |
| Myopia (n = 3) | | | | | | |
| CRx | -2.67 ± 2.40 | 0.32 | 0.83 ± 1.10 | 0.61 | -2.25 ± 1.90 | 0.28 |
| PlusoptiX | -1.04 ± 0.83 | | 0.46 ± 0.44 | | -0.81 ± 0.65 | |
| Difference | 1.63 ± 1.57 | | 0.37 ± 0.66 | | 1.44 ± 1.25 | |

SE: Spherical equivalent; CRx: Cycloplegic Retinoscopy; NA: not applicable.

Difference is absolute cycloplegic minus PlusoptiX values.

and PlusoptiX was only significant for the sphere and SE of the hyperopic population, with a difference of 2.26 ± 1.30D and 1.84 ± 1.40D, respectively (p<0.05 for both). Screening results and cycloplegic refraction were similar for myopia and astigmatism. In addition, the Bland-Altman plot was generated to further assess the agreement in measured spherical equivalent between PlusoptiX and CR for each eye (Fig 1). According to this analysis, the scatters tended to be within the limits of agreement.

## Correlation between refractive errors and risk factors, age, and sex

To assess correlation between each demographic factors and the prevalence of amblyopia risk factors, a bivariate (Pearson) correlation was performed. A statistically significant correlation was seen between systemic disorders (mostly hematologic, neurologic, cardiac and metabolic disorders) and prevalence of amblyopia risk factors with p<0.001; however this correlation was a weak one (Pearson correlation = 0.13). On the other hand, no significant correlation was seen between prematurity and referral in our small cohort (p = 0.86). There was no significant correlation between detection of amblyopia risk factors and age (p = 0.63), nor with gender (p = 0.76). The distribution of refractive errors varied with age, this is demonstrated in Fig 2. Hyperopia was the most commonly encountered refractive error overall especially in the older age group.

## Discussion

This study reported amblyopia risk factors in Lebanese children aged 2 to 6 years in a hospital-based setting using automated vision screening. The referral rate for full eye exam was 5.7%. Out of those examined, 73% needed glasses. The refractive amblyopia risk factors as detected by the PlusoptiX screener were myopia in 0.3% hyperopia in 1.7% and astigmatism in 1.4%. Among the referred and examined subjects, the distribution of refractive errors as confirmed

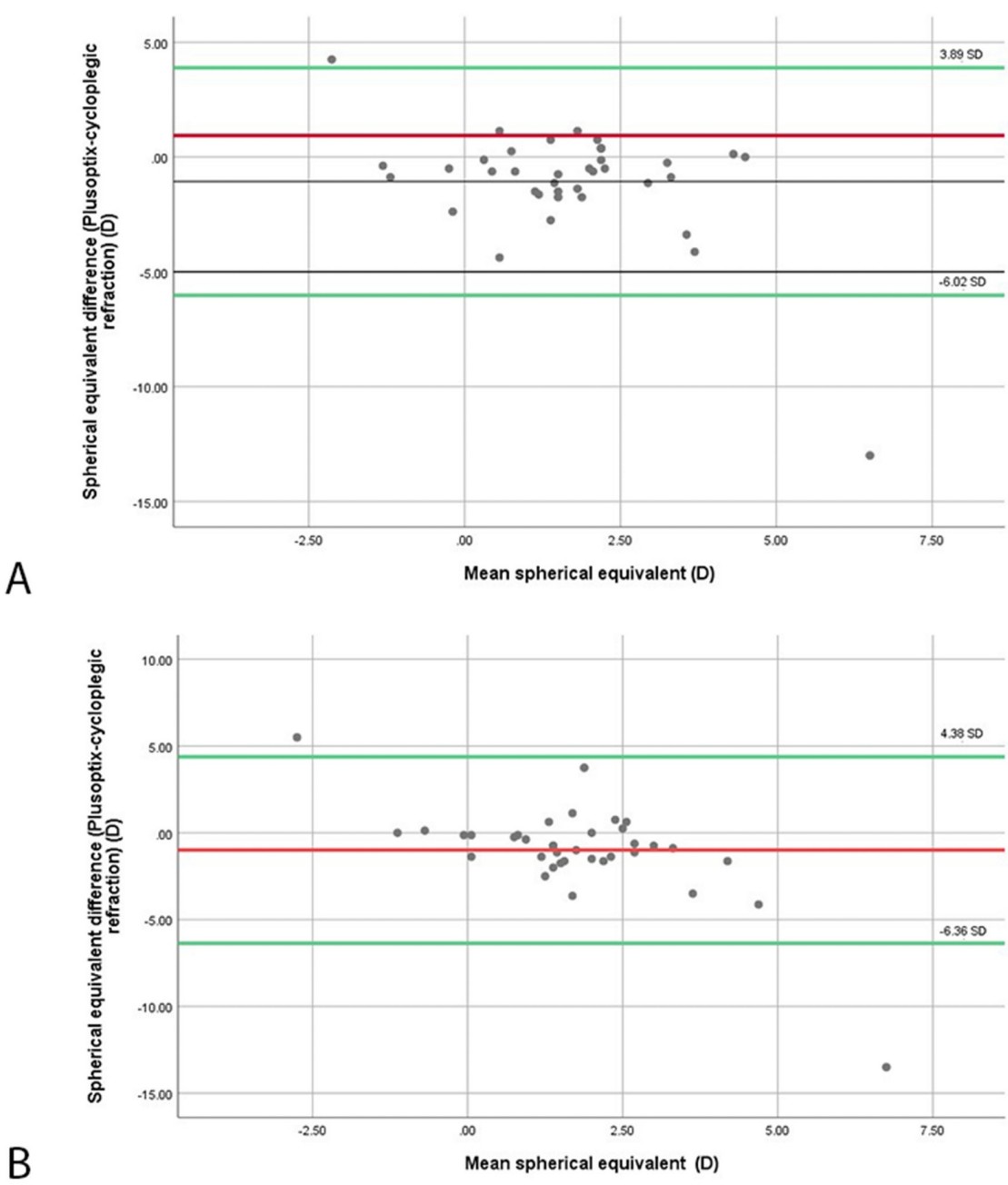

**Fig 1. Bland-Altman plot of spherical equivalent values of the right eye (OD) (A) and the left eye (OS) (B) in diopters.** The difference between the mean spherical equivalent as measured by PlusoptiX and mean spherical equivalent as measured by cycloplegic retinoscopy are plotted on the y-axis, and the mean spherical equivalent value obtained by the two measurements on the x-axis. Dashed lines represent the upper and lower 95% limits of agreement.

by cycloplegic refraction was: 8% myopia, 51% hyperopia and 41% astigmatism. The prevalence of refractive amblyopia risk factors was not significantly correlated with age, gender or prematurity, but with the presence of systemic disorders.

The PlusoptiX vision screener has proven to be sensitive and specific in detecting amblyopia risk factors and refractive errors [3]. Arnold et al reported sensitivity of PlusoptiX at 84%, and its specificity at 94% [18]. A more recent study reviewed 18 papers: the sensitivity ranged

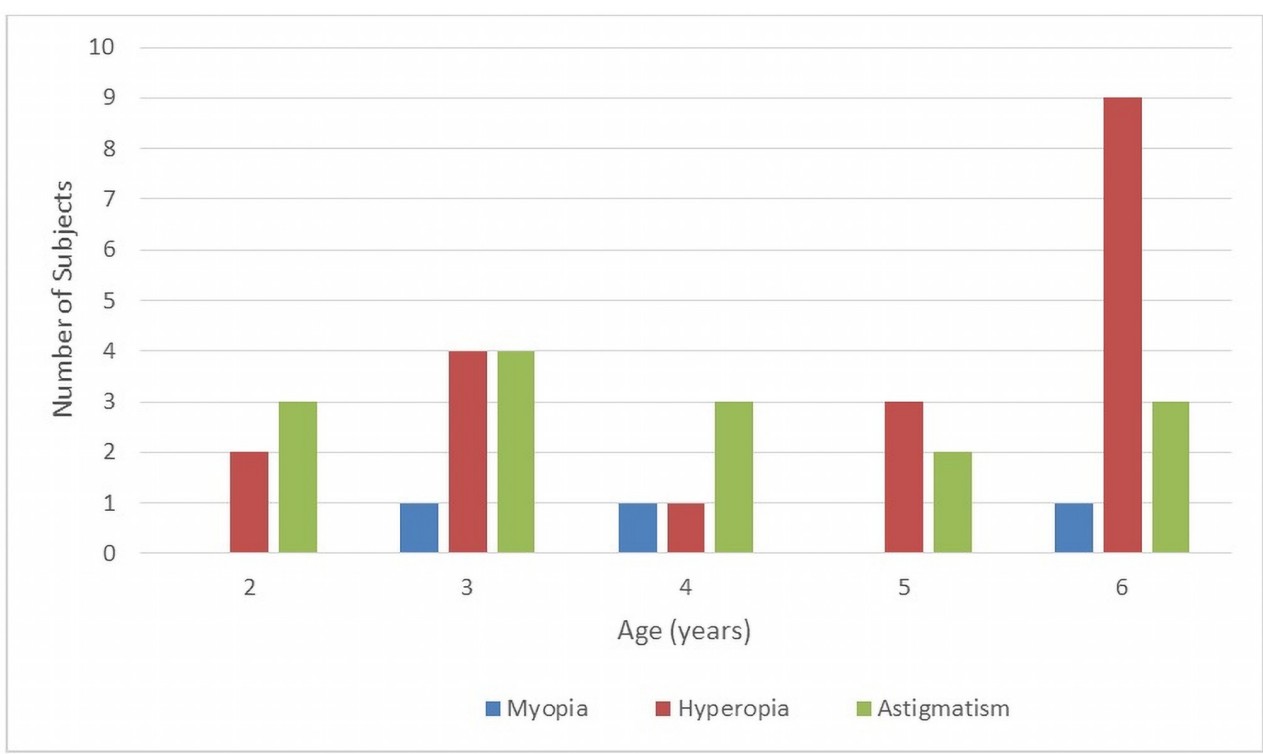

**Fig 2. The distribution of refractive errors according to age in the examined population.**

between 47% and 99%, and specificity from 49% to 100% [27]. The referral rate in our present study was 5.7% of those screened, very close to that determined in a similar study from the United Kingdom (5.6%) [28]. The referral rate in Hawaii using the same screener was however higher (8%), and that reported from China was lower (3.9%) [17, 19].

The prevalence of refractive amblyopia risk factors reported in the literature varies by the cut-off limits followed, method used, region where the study was performed, and age group studied. In Lebanon, only a single previous report looked into amblyopia risk factors in a cohort of 935 school children between the age of 5 and 18 years [29]. Visual acuity was solely tested, and "ametropia" was noted in 15.7% of the screened population, 70% of whom were unaware of their visual defect [29]. The prevalence of ametropia detected in our screened population was lower; however, our age group was narrower, and we had more stringent criteria based on automated vision screening, and not only subjective visual acuity which may be inaccurate in young subjects. An interesting study conducted in Australia looked at a cohort of predominantly Lebanese school children 10 to 15 years of age [30]. They aimed at detecting the prevalence rate of refractive errors among children of Middle Eastern descent (85.3% were Lebanese), raised and living in Australia [30]. This was the only previous study looking at refractive errors in a Lebanese cohort. A non-cycloplegic auto-refraction was conducted for 354 schoolchildren using the autorefractor NVision-K5001: 14.7% of the studied subjects were found to have myopia (sphere $\leq$-0.75D), and 21.8% had hyperopia (sphere $\geq$+1.50D) [30]. These numbers were closer to those of Australian origin than to those reported from the Middle Eastern region. The authors concluded that the prevalence of refractive errors tended to resemble that of the host country, suggesting that lifestyle and the education system had a higher impact on refractive errors than genetic and ethnic background [30]. The prevalence of

refractive amblyopia risk factors in the current study are lower than those detected in this Lebanese cohort living in Melbourne, albeit with a different age group.

In Middle Eastern countries, the reported prevalence rates in general were higher than our current study. In Iranian children aged 7 to 12 years, prevalence rates of 4.9% for myopia, 3.5% for hyperopia and 22.6% for astigmatism were reported [20]. In Saudi Arabia, the prevalence rate of hyperopia was 1.5%, myopia 0.7% and astigmatism 25.3% [31]. In Jordan, children 12 to 17 years of age were screened; referral rate after vision screening was 25%, quite higher than ours at 5.7% [32]. The distribution of errors differed from ours with a higher prevalence of all refractive errors, mainly myopia, as compared to a dominance of hyperopia in our studied population. This can be attributed to the difference in the studied age groups, as hyperopia is more prevalent in younger children while myopia tends to increase with age.

The Vision In Preschoolers (VIP) study screened preschool children using several methods, including autorefractors and non-cycloplegic retinoscopy [33]. Hyperopia was seen in 2.5%, myopia in 0.1% and astigmatism in 4.4% [33] similar to our cohort. Another study in preschool children (3 to 5 years of age) using the Retinomax autorefractor showed a referral rate of 8.9% and 74% of the examined patients needed glasses [34]. In the Baltimore Pediatric Eye Disease Study, myopia was found in 0.7% and hyperopia in 8.9% [35]. In Europe, the prevalence of the different types of refractive errors were higher than ours [36–38]. The prevalence of myopia is higher in the Far East as was demonstrated by the Strabismus, Amblyopia, and Refractive error in young Singaporean children (STARS) study in Singapore [39]. The prevalence rates were 11% for myopia, much higher than our present study, 1.4% for hyperopia, very close to the rate reported in this study and 8.6% for astigmatism, also higher than what we reported [39]. They concluded that change in prevalence may be influenced by ocular development, environment and testability of myopia [39]. A further study in Bhutan revealed a presence of 6.6% for myopia and 2.2% for hyperopia [40]. In India, a systematic review of 12 studies reported that the prevalence of refractive errors in children ≤15 years of age was: 1.4% for myopia, 4% for hyperopia and 1.1% for astigmatism [41]. In Chinese children of similar age to ours, prevalence rates were higher [42].

Our present study showed that there was no significant correlation between rate of referral and neither age nor gender. Hyperopia was the most prevalent refractive error in our examined population. One explanation is that young subjects with hyperopia are usually asymptomatic, and thus hyperopia goes mostly undetected in children. The prevalence of amblyopia risk factors and the distribution of refractive errors in the pediatric population vary widely in the reported literature. This is attributed to the absence of unified cut-off limits used to diagnose the different refractive errors. Add to that the differences in ethnic backgrounds of the study populations, age groups and the instruments/ methodology used. This precluded head-to-head comparison of our present work to those from the US, Europe, Asia and the rest of the Middle East.

This study has several strengths; it was conducted on a large sample size with over 1000 children screened. In addition, it was among the first studies to report refractive amblyopia risk factors in the Lebanese pediatric population. It thus provides an important reference for Lebanese pediatric ophthalmologists, and for international ophthalmologists to compare to. Referred patients underwent a full cycloplegic eye examination by the same pediatric ophthalmologist and the AAPOS Vision Screening Committee guidelines were followed for age-based refractive error targets. This study was conducted for a hospital-based population of Lebanese children. This factor, in addition to the limited sample size, served as a limitation, as the detected rates could not be generalized to the community. Another limitation was the low response rate of referral. Add to that, the PlusoptiX detects hyperopia with less sensitivity than it detects myopia and astigmatism (16). Thus, some false negative results might have been

present but not detected. Also, we focused on a narrow age group within the sensitive period of visual development where the condition is amenable to patch therapy/ refractive adaptation and at an age where amblyopia has not yet become entrenched. The rates of systemic associated disorders among the referred subjects were higher than that in the non-referred group probably reflecting that our population came from a tertiary referral center. A wide difference was observed between the mean spherical equivalent of the photoscreener and cycloplegic refraction explained by the use of cycloplegia uncovering more hyperopia. A future community and school-based similar study including a wider age range is underway to reflect the true prevalence in our country.

In summary, using an automated vision screener in a hospital-based cohort of children aged 2 to 6 years, the prevalence of refractive amblyopia risk factors was 5.7%. Hyperopia was the most common refractive error, followed by astigmatism, then myopia. Age, gender, and prematurity were not found to significantly affect the referral rate, while a weak correlation was seen between systemic disorders and an increased risk of amblyopia risk factors.

## Supporting information

**S1 Data.**
(XLSX)

## Author Contributions

**Conceptualization:** Christiane Al-Haddad, Carl-Joe Mehanna, Karine Ismail.

**Data curation:** Zeinab El Moussawi, Stephanie Hoyeck, Carl-Joe Mehanna, Nasrine Anais El Salloukh, Karine Ismail, Mona Hnaini, Rose-Mary N. Boustany.

**Formal analysis:** Christiane Al-Haddad, Zeinab El Moussawi, Stephanie Hoyeck, Carl-Joe Mehanna.

**Funding acquisition:** Christiane Al-Haddad, Stephanie Hoyeck, Karine Ismail.

**Investigation:** Christiane Al-Haddad, Nasrine Anais El Salloukh, Karine Ismail, Mona Hnaini, Rose-Mary N. Boustany.

**Methodology:** Christiane Al-Haddad, Nasrine Anais El Salloukh, Karine Ismail, Rose-Mary N. Boustany.

**Supervision:** Christiane Al-Haddad, Carl-Joe Mehanna, Nasrine Anais El Salloukh, Rose-Mary N. Boustany.

**Writing – original draft:** Christiane Al-Haddad, Zeinab El Moussawi, Stephanie Hoyeck, Carl-Joe Mehanna, Nasrine Anais El Salloukh.

**Writing – review & editing:** Christiane Al-Haddad, Zeinab El Moussawi, Nasrine Anais El Salloukh, Mona Hnaini, Rose-Mary N. Boustany.

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
