## [Decision Letter · Decision Letter 0]

17 Mar 2021

PONE-D-21-00998

Refractive Errors among Pediatric Patients in a Hospital-Based Setting Using Photoscreening

PLOS ONE

Dear Dr. Al-Haddad,

Thank you for submitting your manuscript to PLOS ONE. After careful consideration, we feel that it has merit but does not fully meet PLOS ONE’s publication criteria as it currently stands. Therefore, we invite you to submit a revised version of the manuscript that addresses the points raised during the review process.

ACADEMIC EDITOR:

The manuscript has many fundamental issues that need meticulous revision.

1- The manuscript needs English editing for grammar and style

2- Some sections such as the introduction section are too long and contain unnecessary data while the results section needs more details

3- The fact that refractive errors were diagnosed with photoscreening rather than a more accurate method like retinoscopy or conventional autorefractors should be addressed. The authors might consider using the term "detection of amblyopia risk factors" rather than "refractive errors"

We look forward to receiving your revised manuscript.

Kind regards,

Ahmed Awadein, MD, Ph.D, FRCS

Academic Editor

PLOS ONE

Journal Requirements:

2) We note that the grant information you provided in the ‘Funding Information’ and ‘Financial Disclosure’ sections do not match.

Reviewers' comments:

Reviewer's Responses to Questions

**Comments to the Author**

1. Is the manuscript technically sound, and do the data support the conclusions?

Reviewer #1: No

Reviewer #2: No

Reviewer #3: Yes

2. Has the statistical analysis been performed appropriately and rigorously? 

Reviewer #1: N/A

Reviewer #2: No

Reviewer #3: Yes

3. Have the authors made all data underlying the findings in their manuscript fully available?

Reviewer #1: No

Reviewer #2: No

Reviewer #3: Yes

4. Is the manuscript presented in an intelligible fashion and written in standard English?

Reviewer #1: No

Reviewer #2: No

Reviewer #3: Yes

5. Review Comments to the Author

Reviewer #1: I have one fundamental objection against methodology of this study: I do not agree that photoscreener can be used for evaluation of refractive error prevalence, and this is also showed by the results of this study. It is however accepted that it is used in screening of amblyopia risk factors. Because of that the whole study, to be accepted for publication, must be completely re-written. Moreover, I do not think that referral rate is so much interesting as long as you did not analyze also all negative cases. However, verification with complete ophthalmic examination and cycloplegia refraction of positive cases is very interesting issue and could a subject for a separate article in the future.

Reviewer #2: This study shows the results of hospital-based screening of Lebanese pediatric patients 2-6 years regarding amblyogenic refractive errors and the results of subsequent comprehensive ophthalmic evaluation. There are some concerns regarding methodology and end results.

1. The manuscript needs significant English language editing.

2. The “INTRODUCTION” section is very confusing and contains lots of unnecessary data. Specifically, the second paragraph should be totally rephrased. The whole “INTRODUCTION” section should be 2 to 3 paragraphs describing in brief background and the aim of the study.

3. The authors did not mention the period of time during which the study was conducted.

4. In “RESULTS” section, the authors mentioned that only 59% of their referred cases were examined at their facility, without mentioning where the rest of referred cases (31%) were examined?

Reviewer #3: A very nice, respectful and valuable work that is worthy more evaluation with a larger number of patients for a better benefit.

29 prevalence (would be better than rate)

73-76 If you would like to rephrase.

129-133 Table 1 AAPOS ARFs (if you would like to mention ; limit of astigmatism 90/180 Vs oblique one ?, type of anisometropia)

140 Was CRx performed by the same examiner ?

141,142 Source of this rational of cycloplegia ?

174 37/63 were examined at your department. What about the remainder ?

312 "still be treated" Do you mean amblyopia ttt with refractive adaptation only or in general ?

6. PLOS authors have the option to publish the peer review history of their article (what does this mean?). If published, this will include your full peer review and any attached files.

Reviewer #1: No

Reviewer #2: No

Reviewer #3: No

---

## [Author Response · Author response to Decision Letter 0]

21 Apr 2021

ACADEMIC EDITOR:

The manuscript has many fundamental issues that need meticulous revision.

1- The manuscript needs English editing for grammar and style

We have revised thoroughly the whole manuscript for grammatical and linguistic errors/ style. 

2- Some sections such as the introduction section are too long and contain unnecessary data while the results section needs more details

The introduction was made more succinct and only relevant data were kept. The results section was elaborated more on. 

3- The fact that refractive errors were diagnosed with photoscreening rather than a more accurate method like retinoscopy or conventional autorefractors should be addressed. The authors might consider using the term "detection of amblyopia risk factors" rather than "refractive errors"

Thank you for the comment, the authors agree and have modified the title and manuscript text to use “amblyopia risk factors” rather than “refractive errors.”

Reviewer #1: I have one fundamental objection against methodology of this study: I do not agree that photoscreener can be used for evaluation of refractive error prevalence, and this is also showed by the results of this study. It is however accepted that it is used in screening of amblyopia risk factors. Because of that the whole study, to be accepted for publication, must be completely re-written. Moreover, I do not think that referral rate is so much interesting as long as you did not analyze also all negative cases. However, verification with complete ophthalmic examination and cycloplegia refraction of positive cases is very interesting issue and could a subject for a separate article in the future.

We thank the reviewer. We have modified accordingly the title to “Amblyopia Risk Factors among Pediatric Patients in a Hospital-Based Setting Using Photoscreening;” additionally, we revised the manuscript text to reflect our aim as detection of amblyopia risk factors in children visiting our medical center rather than prevalence of refractive errors. We agree that reporting the validation of photoscreening positive results by cycloplegic refraction will be an interesting topic in a future study. 

Reviewer #2: This study shows the results of hospital-based screening of Lebanese pediatric patients 2-6 years regarding amblyogenic refractive errors and the results of subsequent comprehensive ophthalmic evaluation. There are some concerns regarding methodology and end results.

1. The manuscript needs significant English language editing.

We have revised the whole manuscript for grammatical and linguistic errors. 

2. The “INTRODUCTION” section is very confusing and contains lots of unnecessary data. Specifically, the second paragraph should be totally rephrased. The whole “INTRODUCTION” section should be 2 to 3 paragraphs describing in brief background and the aim of the study.

The introduction was made more succinct and relevant points were kept. Specifically paragraph 2 was rephrased and the whole introduction was shortened. 

3. The authors did not mention the period of time during which the study was conducted.

The study was conducted over a time span of 2 years (2018-2020), this was added to the methods section. 

4. In “RESULTS” section, the authors mentioned that only 59% of their referred cases were examined at their facility, without mentioning where the rest of referred cases (31%) were examined?

The rest of referred cases were lost to follow up after multiple attempts to schedule an ophthalmology appointment. This was now added to the text for clarification.

Reviewer #3: A very nice, respectful and valuable work that is worthy more evaluation with a larger number of patients for a better benefit.

We thank the reviewer for their positive feedback and appreciate their comment, indeed we plan on a larger scale “school-based” prevalence study when time permits.

29 prevalence (would be better than rate)

Thank you, it was corrected accordingly.

73-76 If you would like to rephrase.

The section was rephrased for better clarity to the reader.

129-133 Table 1 AAPOS ARFs (if you would like to mention ; limit of astigmatism 90/180 Vs oblique one ?, type of anisometropia)

Although it has been reported that oblique astigmatism constitutes a greater amblyopia risk factor than regular astigmatism, the guidelines published by the AAPOS Vision Screening Committee (JAAPOS 2013;17:4-8, Table 1: AAPOS ARFs), which we have followed in this study, did not differentiate by the axis of astigmatism, but rather used the magnitude in Diopters. Similarly, for anisometropia, only the absolute value in diopters was used. 

140 Was CRx performed by the same examiner?

Yes the same pediatric ophthalmologist (author C Al-Haddad) examined all referred patients, this is now mentioned explicitly in the methods section.

141,142 Source of this rational of cycloplegia ?

Thank you for your comment. In this clinic, we use two sets of dilating drops as a standard especially with darker irides (which constitute a good portion of our population, below references). This was added to our methods section. 

References : 

Farhood, Q. K. (2012). Cycloplegic refraction in children with cyclopentolate versus atropine. J Clin Exp Ophthalmol, 3(7), 1-6.

Wallace, D. K., Morse, C. L., Melia, M., Sprunger, D. T., Repka, M. X., Lee, K. A., & Christiansen, S. P. (2018). Pediatric eye evaluations preferred practice Pattern®: I. vision screening in the primary care and community setting; II. comprehensive ophthalmic examination. Ophthalmology, 125(1), P184-P227.

174 37/63 were examined at your department. What about the remainder ?

The rest of referred cases were lost to follow up after multiple attempts to schedule an ophthalmology appointment. This was now added to the text.

312 "still be treated" Do you mean amblyopia ttt with refractive adaptation only or in general ?

This was clarified in text: “within the sensitive period of visual development, while still amenable to patch therapy/ refractive adaptation and at an age where amblyopia has not yet become entrenched”

---

## [Decision Letter · Decision Letter 1]

2 Jun 2021

PONE-D-21-00998R1

Amblyopia Risk Factors among Pediatric Patients in a Hospital-Based Setting Using Photoscreening

PLOS ONE

Dear Dr. Al-Haddad,

Thank you for submitting your manuscript to PLOS ONE. After careful consideration, we feel that it has merit but does not fully meet PLOS ONE’s publication criteria as it currently stands. Therefore, we invite you to submit a revised version of the manuscript that addresses the points raised during the review process.

ACADEMIC EDITOR:

Most of the comments were properly addressed. There are only few comments mentioned below that should be addressed.

We look forward to receiving your revised manuscript.

Kind regards,

Ahmed Awadein, MD, Ph.D, FRCS

Academic Editor

PLOS ONE

Journal Requirements:

Reviewers' comments:

Reviewer's Responses to Questions

**Comments to the Author**

1. If the authors have adequately addressed your comments raised in a previous round of review and you feel that this manuscript is now acceptable for publication, you may indicate that here to bypass the “Comments to the Author” section, enter your conflict of interest statement in the “Confidential to Editor” section, and submit your "Accept" recommendation.

Reviewer #1: All comments have been addressed

Reviewer #3: All comments have been addressed

Reviewer #4: All comments have been addressed

2. Is the manuscript technically sound, and do the data support the conclusions?

Reviewer #1: Yes

Reviewer #3: Yes

Reviewer #4: Yes

3. Has the statistical analysis been performed appropriately and rigorously? 

Reviewer #1: Yes

Reviewer #3: Yes

Reviewer #4: Yes

4. Have the authors made all data underlying the findings in their manuscript fully available?

Reviewer #1: Yes

Reviewer #3: Yes

Reviewer #4: Yes

5. Is the manuscript presented in an intelligible fashion and written in standard English?

Reviewer #1: Yes

Reviewer #3: Yes

Reviewer #4: Yes

6. Review Comments to the Author

Reviewer #1: Thank you for revising the ms that is now significantly improved.

Reviewer #3: (No Response)

Reviewer #4: 1- Abstract: Results "The refractive errors among the referred patients were: 41% astigmatism, 51% hyperopia, and 8% myopia, with no amblyopia detected on eye examination.: Needs rephrasing

2- Add reference to Table 1

3- Methods: “for verbal children; anterior segment examination; motility examination for any eye misalignment; cycloplegic manual retinoscopy to detect refractive errors (30 minutes after pupillary dilation with Mydriacyl 1% and Cyclopentolate 1%, applied twice 10 minutes apart, in this clinic, we use two sets as a standard especially with darker irides); The sentence needs grammatical revision. It also starts with small letter

4- Results: "Out of the 37 examined children, 19 had hyperopia, 15 had astigmatism and 3 subjects were myopic." So do this means that none of the patients had both spherical and cylindrical error at the same time?

5- While correlation was done to compare the readings of autorefractor and retinoscopy, the correct statistical analysis would to use an agreement statistics e.g. Bland-Altman plot and Deming regression analysis

7. PLOS authors have the option to publish the peer review history of their article (what does this mean?). If published, this will include your full peer review and any attached files.

Reviewer #1: **Yes: **Andrzej Grzybowski, MD, PhD

Reviewer #3: **Yes: **Ahmed Taha Ismail

Reviewer #4: No

---

## [Author Response · Author response to Decision Letter 1]

28 Jun 2021

June 28, 2021

Dear Dr Chenette and PLOS ONE Editorial Board,

Thank you for your response to our revision on our manuscript entitled "Amblyopia Risk Factors among Pediatric Patients in a Hospital-Based Setting Using Photoscreening". We appreciate the time and effort that has gone into reviewing our manuscript and thank you for the valuable comments.

Please find below a point-by-point reply to the reviewer comments. Comments are copied verbatim, followed immediately by our response. 

Reviewer #4:

1- Abstract: Results "The refractive errors among the referred patients were: 41% astigmatism, 51% hyperopia, and 8% myopia, with no amblyopia detected on eye examination.: Needs rephrasing

This sentence was rephrased as follows: 

The refractive errors observed among the examined patients were distributed as follows: 41% astigmatism, 51% hyperopia, and 8% myopia; amblyopia was not detected.

2- Add reference to Table 1

 Reference was added to the table title. 

3- Methods: “for verbal children; anterior segment examination; motility examination for any eye misalignment; cycloplegic manual retinoscopy to detect refractive errors (30 minutes after pupillary dilation with Mydriacyl 1% and Cyclopentolate 1%, applied twice 10 minutes apart, in this clinic, we use two sets as a standard especially with darker irides); The sentence needs grammatical revision. It also starts with small letter

 We agree that this was a long run-on sentence. It was reworded as follows:

This entailed age-dependent visual acuity testing: “central steady & maintained” testing for preverbal children and vision charts (Allen pictures and Early Treatment Diabetic Retinopathy Study charts (EDTRS)) for verbal children. Anterior segment examination, motility examination for any eye misalignment, and posterior segment examination using indirect ophthalmoscopy were performed. Cycloplegic manual retinoscopy was done to detect refractive errors (30 minutes after pupillary dilation with Mydriacyl 1% and Cyclopentolate 1%, applied twice 10 minutes apart; in this clinic, we use two sets as a standard especially with darker irides).

4- Results: "Out of the 37 examined children, 19 had hyperopia, 15 had astigmatism and 3 subjects were myopic." So do this means that none of the patients had both spherical and cylindrical error at the same time?

Thank you for your comment. As we mentioned in the Methods section:

“Significant astigmatism was defined as cylindrical power of 1D or more with the main refractive error being the cylinder. Those reported with hyperopia and myopia had mainly spherical refractive error (>+1D or <-1D, respectively).”

Thus, the patients were subdivided based on the predominant refractive error detected when both spherical and cylindrical errors co-existed. 

5- While correlation was done to compare the readings of autorefractor and retinoscopy, the correct statistical analysis would to use an agreement statistics e.g. Bland-Altman plot and Deming regression analysis

Thank you for your comment. We agree that an agreement statistic would strengthen our study. We thus added a Bland-Altman plot to our paper.

---

## [Editor Report · Decision Letter 2]

5 Jul 2021

Amblyopia Risk Factors among Pediatric Patients in a Hospital-Based Setting Using Photoscreening

PONE-D-21-00998R2

Dear Dr. Al-Haddad,

We’re pleased to inform you that your manuscript has been judged scientifically suitable for publication and will be formally accepted for publication once it meets all outstanding technical requirements.

Kind regards,

Ahmed Awadein, MD, Ph.D, FRCS

Academic Editor

PLOS ONE
---

## [Editor Report · Acceptance letter]

12 Jul 2021

PONE-D-21-00998R2 

Amblyopia Risk Factors among Pediatric Patients in a Hospital-Based Setting Using Photoscreening 

Dear Dr. Al-Haddad:

I'm pleased to inform you that your manuscript has been deemed suitable for publication in PLOS ONE. Congratulations! Your manuscript is now with our production department. 

Kind regards, 

on behalf of

Dr. Ahmed Awadein 

Academic Editor

PLOS ONE